# Mental Fatigue, But Not other Fatigue Characteristics, as a Candidate Feature of Obsessive Compulsive Personality Disorder in Patients with Anxiety and Mood Disorders—An Exploratory Study

**DOI:** 10.3390/ijerph17218132

**Published:** 2020-11-03

**Authors:** Julija Gecaite-Stonciene, Naomi A. Fineberg, Aurelija Podlipskyte, Julius Neverauskas, Alicja Juskiene, Narseta Mickuviene, Julius Burkauskas

**Affiliations:** 1Laboratory of Behavioral Medicine, Neuroscience Institute, Lithuanian University of Health Sciences, LT-00135 Palanga, Lithuania; aurelija.podlipskyte@lsmuni.lt (A.P.); julius.neverauskas@lsmuni.lt (J.N.); alicja.juskiene@lsmuni.lt (A.J.); narseta.mickuviene@lsmuni.lt (N.M.); julius.burkauskas@lsmuni.lt (J.B.); 2National Obsessive Compulsive Disorders Specialist Service, Hertfordshire Partnership University, NHS Foundation Trust, Welwyn Garden City AL8 6HG, UK; naomi.fineberg@btinternet.com

**Keywords:** obsessive compulsive personality, fatigue, mental exertion, cognitive/mental demand, executive function, mood disorder, anxiety disorder

## Abstract

Background: Obsessive compulsive personality disorder (OCPD) is commonly associated with anxiety and mood disorders (AMDs), in which fatigue and executive dysfunction represent key symptoms. Executive dysfunction has also been demonstrated in subjects with OCPD, and is additionally found to be a cardinal feature of fatigue. This study aimed to investigate the associations between fatigue, executive dysfunction, and OCPD in patients with AMDs. Methods: In this cross-sectional study, 85 AMD patients (78% females, mean age 39 ± 11 years) were evaluated for OCPD traits by using the observer-rated Compulsive Personality Assessment Scale. The Multidimensional Fatigue Inventory-20 was used to measure different aspects of fatigue, and the Trail Making Test was employed to assess executive functioning. The Hamilton rating scales were used to evaluate anxiety and depression symptoms. Results: Controlling for potential confounders, there was a significant link between OCPD and mental fatigue (OR, 1.27; 95% CI, 1.02 to 1.58; *p* = 0.033). No associations were found between the presence of OCPD and other relevant fatigue characteristics, including general fatigue, physical fatigue, reduced activity, and reduced motivation, as well as executive functions. Conclusions: To the best of our knowledge, this study is the first to report associations between OCPD and mental fatigue in patients with AMDs, suggesting mental fatigue as a clinically important symptom when considering particular personality pathologies.

## 1. Introduction

Obsessive compulsive personality disorder (OCPD) is characterized by a chronic maladaptive pattern of excessive perfectionism, preoccupation with orderliness and detail, cognitive rigidity, and need for control [1,2]. It represents one of the most prevalent personality disorders, affecting almost 8% of the general population worldwide [1,3]. OCPD is highly prevalent in various psychiatric populations [4], including patients with anxiety and mood disorders (AMDs), where the prevalence is reported to increase to around 25% [5,6,7,8,9], and is thought to signify a poor clinical outcome after antidepressant treatment [10]. Yet, OCPD is still under-researched [11] and frequently under-diagnosed, even in clinical populations [12]. To date, no evidence-based treatments have been established, though a possible therapeutic role for selective serotonin inhibitors or cognitive remediation therapies has been proposed [13,14].

Perfectionism, a core symptom of OCPD, represents one of the key factors involved in the development and maintenance of anxiety and depressive symptoms [15,16], as well as chronic fatigue [17]. In fact, traits of OCPD are commonly associated with anxiety and depression [18], disorders of which include fatigue as an intrinsic element.

Furthermore, several authors analyzing separate traits that have been attributed to OCPD, such as workaholism [19,20,21] and perfectionism [8,17], suggest their possible association with fatigue. Due to the tendency of “workaholic behavior” [12,21], people with OCPD are more prone to work-related fatigue, which itself is associated with work-related rumination and poorer sleep quality [22]. Furthermore, the perfectionism trait, which is part of OCPD conceptualization, is considered a pre-disposing factor to persistent fatigue [17] due to increased mental activity and excessively high standards [23]. Taken together, the lack of recovery after work-related stress, together with excessive perfectionism, may lead a person with OCPD to depletion of emotional resources, which may contribute to developing chronic fatigue [24]. However, so far, no study has investigated the possibility of a specific association between fatigue and OCPD.

It is well established that AMDs are associated with a broad spectrum of neurocognitive changes, including significant executive dysfunction [25,26,27,28,29,30,31,32]. A meta-analysis by McDermott et al. [25] suggested that increased severity of depression is linked with reduced cognitive performance, specifically in the areas of episodic memory, executive functions, and processing speed. Another systematic review by Lee et al. [27] reported poor neuropsychological performance in those with first-episode major depressive disorder. Cognitive deficits are also present in patients with bipolar disorder, as meta-analyzed by Depp et al. [28]. Similarly, cognitive impairment has also been found in those with generalized anxiety disorder [26].

Nevertheless, the neurocognitive mechanisms underpinning OCPD are still under-researched. In a recent study, executive dysfunction, particularly cognitive inflexibility (inflexible set shifting) and impaired executive planning, was demonstrated in a group of subjects with OCPD [1] in a non-clinical sample, excluding those with AMDs. Prior research has also reported executive dysfunction in patients with obsessive compulsive disorder [33,34,35], which is a different yet often comorbid disorder with OCPD, sharing conceptual and nosological similarities [12,36]. Prior research has also reported executive dysfunction, specifically inflexible task switching (a component in set shifting), as a cardinal feature of mental fatigue in a post-myocardial infarction population [37]. This was the first study to clearly indicate an association between mental fatigue and executive functioning in this population [38].

Taken together, these findings suggest that a specific association may exist between OCPD, executive impairment, and fatigue in patients with AMD that has thus far not been explored.

In this study, we aimed to investigate whether measures of mental fatigue and cognitive functioning are associated with the presence of OCPD in a sample of patients with AMD. If such a relationship were to be reliably demonstrated, the findings would be of clinical value, as clinicians would be alerted to look for the presence of OCPD in patients presenting with AMDs characterized by excessive fatigue, and could adapt the care plan accordingly.

## 2. Materials and Methods

### 2.1. Participants

This cross-sectional case–control study included 91 consecutive AMD patients attending an outpatient stress disorder unit at the Palanga Clinic of the Neuroscience Institute, Lithuanian University of Health Sciences. Six patients (7%) were unwilling to participate in the study. The final sample consisted of 85 patients (78% females and 22% males) with a mean age of 39 ± 11 years. All study patients had comorbid AMD diagnoses as follows: Major depressive episode, 71 patients (84%); dysthymia, 14 patients (17%); bipolar disorder, 5 patients (6%); generalized anxiety disorder, 37 patients (44%); panic disorder, 37 patients (44%); agoraphobia, 53 patients (62%); and social anxiety disorder, 30 patients (35%) (Table 1). All patients received standard treatments for their AMDs according to their clinical needs. The commonest medications used were selective serotonin reuptake inhibitors (SSRIs), benzodiazepines, antipsychotics, and mirtazapine, taken either as a monotherapy or in various combinations. The amount of medication used was similar in both groups; 25% of the OCPD group were not taking any medication compared to 18.5% in the control group (*p* = 0.524) and there was no significant overuse of any one type of medication or medication combination in the OCPD group compared to the controls (see Appendix A). All of the patients received psychological–psychotherapeutic treatment according to their problems, including individual psychological counseling, group psychotherapy, dance/movement therapy, art therapy, and relaxation training.

### 2.2. Treatment Setting and Procedures

Sociodemographic, clinical, and psychological characteristics were evaluated during the first three days of admission to the unit. Patients also completed a questionnaire, assessing subjectively experienced fatigue symptoms and completed tests measuring cognitive functioning. The Lithuanian Biomedical Research Ethics Committee (project identification code: Stresogen no.1, 25 February 2019) approved the study protocol and each study participant gave informed consent before inclusion into the study. The study was conducted in accordance with the Declaration of Helsinki.

### 2.3. Measures

Sociodemographic and clinical information, including diagnosis (as defined by The Mini International Neuropsychiatric Interview) [38], medication use, and history of smoking, were collected.

The presence and severity of OCPD traits were evaluated using the observer-rated Compulsive Personality Assessment Scale (CPAS) [39]. Based on a semi-structured interview, each of the eight Diagnostic Statistical Manual-5 (DSM-5) diagnostic criteria is scored on a scale of zero to four, and the maximum total score of the CPAS is 32. Consistent with the DSM-5, we operationally defined a diagnosis of OCPD as a score of three (severe) or four (very severe) on at least four of the CPAS items. CPAS had acceptable internal reliability (Cronbach’s α = 0.75).

The Multidimensional Fatigue Inventory-20 (MFI-20) was used to assess general, physical, and mental fatigue, reduced activity, and motivation [40,41]. Patients were asked about their subjectively experienced fatigue during the past few days, rating the severity of symptoms from one (no fatigue) to five (very fatigued). A higher overall score in each sub-scale indicates higher levels of fatigue or reduced motivation. The MFI-20 had an adequate internal reliability (Cronbach’s α = 0.91).

The 14-item Hamilton Anxiety Scale (HAM-A) and the 17-item Hamilton Depression Scale (HAM-D) were employed to measure symptoms of anxiety and depression [42,43,44]. The HAM-A (Cronbach’s α = 0.86) and HAM-D (Cronbach’s α = 0.78) had adequate internal reliability.

The Trail Making Test parts A (TMTA) and B (TMTB) [45] measure cognitive processing speed, flexible task switching, and executive control. In TMTA, numbers are presented in a random array and participants have to connect these numbers in numerical order. In TMTB, participants are asked to connect numbers and letters alternately in sequential (numeric and alphabetical) order. The total time in seconds (s) for TMTA reflects cognitive processing speed, while TMTB measures task switching ability. A derivative score for executive control is calculated by taking the scores of TMTA and subtracting them from those of TMTB. The lower the TMTB–A score, the greater the participant’s executive control ability.

### 2.4. Statistical Analyses

Statistical analyses were conducted using version 17.0 of the SPSS for Windows statistical package (SPSS Inc, Chicago, IL, USA). Two-tailed Student’s *t*-tests or Fisher’s χ^2^ tests were applied to compare fatigue and cognitive function, as well as sociodemographic and clinical characteristics in AMD patients with and without OCPD. While exploring the possible associations between fatigue characteristics, executive functioning, and OCPD, we used Benjamini–Hochberg adjustment for multiple comparisons in relevant domains, setting a critical value for a false discovery rate of 0.15 (*p* = 0.03).

For the remaining associations, binary logistic regression analyses were performed to test associations between OCPD, fatigue, and cognitive function, while controlling for possible confounders, including symptoms of anxiety and depression, age, gender, education, medication use, and smoking behavior.

## 3. Results

Baseline characteristics and cross-sectional case–control metrics are presented in Table 1. The average Hamilton Anxiety Scale score was 26 ± 9, representing moderate anxiety [42], while the average Hamilton Depression Scale score was 15 ± 7, suggesting moderate depression [43,44]. Almost one quarter (24%) of the AMD patients fulfilled the operational criteria for OCPD.

Our results show that the level of mental fatigue (though not other forms of fatigue) was higher among OCPD patients than non-OCPD patients (−1.63; 95% CI, −3.08 to −0.17; *p* = 0.029). However, there were no significant differences in terms of executive dysfunction in those with and without the presence of OCPD features (all *p*-values ≥0.05) (Table 2).

The scores on the sociodemographic parameters and clinical characteristics did not show any difference between two groups (all *p*-values ≥ 0.05).

After controlling for the symptoms of anxiety and depression, age, gender, education, medication use, and smoking, mental fatigue remained significantly associated with OCPD diagnosis risk (OR, 1.27; 95% CI, 1.02 to 1.58; *p* = 0.033).

## 4. Discussion

Our exploratory analysis revealed that the presence of OCPD was significantly associated with higher mental fatigue, but not other fatigue characteristics, in AMD patients, even after controlling for possible covariates. However, executive control measures, reflecting processing speed and task switching ability, were not significantly linked with the presence of OCPD.

Our findings show that after adjustment for possible confounding factors, including symptoms of anxiety and depression, age, gender, education, medication use, and smoking, mental fatigue is linked with the presence of OCPD, which is in line with previously published studies in various populations [46,47]. However, the presence of OCPD was not associated with other fatigue characteristics such as general fatigue, physical fatigue, reduced activity, and reduced motivation. In 2015, Calvo and colleagues suggested that personality disorders, including OCPD, which was found in 48.5% of the study sample, were more common in participants with chronic fatigue syndrome than in individuals with no fatigue [24]. In contrast, other fatigue features, including general fatigue and reduced motivation, might be symptoms influenced more by former Axis I disorders such as depression or generalized anxiety disorder. It should be taken into account that in *t*-test comparisons, there was only a trend for statistical significance of higher mental fatigue score in a group of individuals with OCPD vs. a group of individuals without OCPD. However, this association remained after controlling the results for other possible confounding factors. Thus, the presence of mental fatigue might be a unique feature of individuals with OCPD. This reasoning was partially confirmed in a study by Ruiter et al. (2012), where the presence of OCPD was also related to greater mental fatigue severity in 84 adults with chronic insomnia [46].

Mental fatigue is described as a psychobiological state influenced by prolonged periods of demanding cognitive activity [48]. In our study, mental fatigue was assessed using an MFI-20 subscale of four items—“When I am doing something, I can keep my thoughts on it,” “I can concentrate well,” “It takes a lot of effort to concentrate on things,” and “My thoughts easily wander”—mainly reflecting poor concentration and effort needed to maintain focus.

Interestingly, another study conducted by Pasquini and colleagues [47] reported that only when OCPD was present did patients with obsessive compulsive disorder claim to experience higher mental fatigue in contrast to healthy controls. According to Calvo and colleagues, individuals with personality disorders usually demonstrate maladaptive response to negative psychosocial stimuli, which may lead to chronic mental stress [24]. Thus, OCPD may act as a predisposing factor to mental fatigue.

Another explanation may stem from the neurocognitive characteristics of OCPD, as extreme perfectionism and cognitive rigidity, leading to the inflexible pursuit of high standards, the need to exert mental and situational control, as well as workaholism and executive responsibility, may lead a person to overestimate their mental capabilities and exhaust themselves. In fact, these are also characteristics often attributed to patients with chronic fatigue syndrome [24].

Furthermore, as mentioned earlier, the current findings do not show significant links between cognitive functioning and the presence of OCPD in patients with AMDs. The results differed from a previously published study by Fineberg et al. [1] in those without AMDs. This recent study in a student sample found that 21 non-clinical subjects with OCPD were significantly poorer in executive planning as measured using the Stockings of Cambridge task [49] and the cognitive flexibility task as measured on the intradimensional/extradimensional set shift, in contrast to 15 healthy controls without OCPD [1]. However, on the contrary, in another study of 24 patients with traumatic brain injury [50], executive functioning, as measured by the Wisconsin card sorting test, was not associated with the presence of OCPD. The latter findings showing non-significant links between these two variables are in line with our results.

Several explanations might underlie these inconsistent findings, such as methodological differences (i.e., the use of different measures of cognitive function) and relatively small sample sizes across all of the studies. The findings may also differ due to the distinct clinical characteristics ranging from non-clinical subjects to patients with traumatic brain injury. As with fatigue, executive functioning is known to be affected by anxiety and depression [51]. In our study, in particular, the presence of anxiety and depression may have obscured any additional effect of OCPD on cognitive functioning. Thus, the effect of the former Axis I disorders on executive functioning might be stronger than the effect of personality traits. Therefore, these associations might be more apparent in studies comparing the cognitive functioning of healthy controls vs. individuals with OCPD [51].

In contrast to Fineberg et al. [1], we used the TMT, which is considered to be a relatively shorter and easier cognitive test, and might not be sensitive enough to reflect the complex neurocognitive mechanisms of executive dysfunction in OCPD. Patients with OCPD, due to their conscientiousness and greater concern for details, might have been better able to complete a test not requiring complex cognitive skills in a short period of time. However, to confirm this assumption, further studies using various cognitive function instruments are required.

Although these associations were found to be independent of major confounding factors, the exact mechanisms linking mental fatigue with OCPD traits are still unknown. Thus, future research should explore the possible role of mediating factors in this association, including the role of executive functioning. A larger clinical sample and more complex neurocognitive measures are also needed to make more accurate predictions. Further, investigation of mental fatigue in diverse groups of patients is also recommended to confirm the current finding.

There are several limitations noteworthy of mention in this report. First, since this study was based on a cross-sectional design, assumptions of causality must be made with caution. Second, patients were recruited from a single clinic, which may have introduced selection bias into the results. It is also worth mentioning that the number of subjects was relatively small. This was an exploratory study where we used the Benjamini–Hochberg criterion, which is considered to be a liberal adjustment method for correcting for multiple comparisons. However, in particular, this criterion allows to set a false discovery rate, controlling for the expected proportion of significant findings that are false and is a recommended alternative to Bonferroni-type adjustments in health studies [52]. Future research should focus on obtaining a larger sample population of AMD patients in order to confirm the findings. Finally, in the DSM-5, the presence of OCPD can be determined based on the classical as well as an alternative models for personality disorders (AMPDs). In this study, we used the CPAS scale, which is based on the classical DSM-5 OCPD diagnostic approach. It was beyond our study scope to measure OCPD within an AMPD framework. However, this limitation can be addressed in future studies.

These results suggest an important first step toward better understanding the links between OCPD, executive functioning, and mental fatigue in patients with AMDs. However, the mechanism by which these variables may interact still remains to be established with any degree of certainty in future studies.

## 5. Conclusions

Overall, this study demonstrates that in patients with AMDs, mental fatigue, but not other fatigue characteristics or executive functioning, is independently associated with the presence of OCPD. It is also a first study to report that mental fatigue possibly represents a previously under-explored marker of OCPD. Future research on this topic will help to further define the OCPD psychological profile, which will assist with diagnosis and treatment. A greater understanding of our findings could lead to a theoretical improvement in OCPD-related research, which is currently still under development.

## Figures and Tables

**Table 1 ijerph-17-08132-t001:** Baseline characteristics and cross-sectional clinical and cognitive measurements for the study participants.

	Total (*n* = 85)
Age, mean ± SD	39.3 ± 11.2
Gender, *n* (%)	
Men	19 (22.4)
Women	66 (77.6)
Education, *n* (%)	
Tertiary education	24 (28.2)
College/university degree	61 (71.8)
Current medication use, *n* (%)	
Antidepressants	59 (69.4)
Tranquilizers	34 (40.0)
Mood stabilizers	5 (6.0)
Neuroleptics	19 (22.4)
History of smoking, *n* (%)	19 (22.4)
Anxiety score as measured with the Hamilton Anxiety Rating Scale, mean ± SD	25.6 ± 9.4
Depression score as measured with the Hamilton Depression Rating Scale, mean ± SD	15.4 ± 6.8
Trail Making Test, mean ± SD	
Test part A time (s) (cognitive processing speed)	26.8 ± 9.0
Test part B time (task switching)	67.5 ± 23.0
Derivative score test part B–test part A (s) (executive control)	40.7 ± 19.8
Fatigue as measured with the Multidimensional Fatigue Inventory-20, mean ± SD	
General fatigue	16.2 ± 3.4
Physical fatigue	15.5 ± 3.7
Reduced activity	15.4 ± 4.1
Reduced motivation	14.6 ± 3.4
Mental fatigue	15.9 ± 3.8

Notes: Values are given in numbers (%) or mean ± standard deviation (SD).

**Table 2 ijerph-17-08132-t002:** Comparison of cognitive functioning and fatigue levels in patient groups with and without obsessive compulsive personality disorder (OCPD) *.

	Mean Difference (95% CI)	*p*-Value
Trail Making Test		
Test part A time (s) (cognitive processing speed)	−0.89 (−5.50; 3.73)	0.704
Test part B time (task switching)	6.19 (−5,52; 17.89)	0.296
Derivative score test part B–test part A (s)	7.07 (−2.97; 17.11)	0.165
(executive control)		
Fatigue as measured with Multidimensional Fatigue Inventory-20		
General fatigue	−1.08 (−2.78; 0.63)	0.213
Physical fatigue	−0.97 (−2.86; 0.92)	0.311
Reduced activity	−1.15 (−3.22; 0.93)	0.275
Reduced motivation	−1.70 (−3.39; −0.00)	0.050
Mental fatigue	−1.63 (−3.08; −0.17)	0.029

Notes: * Defined operationally using the Compulsive Personality Assessment Scale; CI, confidence interval. *p*-Value calculated by using Student’s *t*-tests for continuous variables.

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
