# Peer review of "Mental Fatigue, But Not other Fatigue Characteristics, as a Candidate Feature of Obsessive Compulsive Personality Disorder in Patients with Anxiety and Mood Disorders—An Exploratory Study"

_ijerph, 2020, doi:10.3390/ijerph17218132_

Round 1
Reviewer 1 Report
This paper reports a cross-sectional study of scores on the Compulsive Personality Assessment Scale and the mental fatigue subsection of the Multidimensional Fatigue Inventory-20. The authors find a correlation between the two. The paper is well-written and the methodology is largely sound (see below for specific comments). Overall, the findings present a very incremental advance in the field and their clinical significance for the general psychiatrist is unclear. Nevertheless, sub-specialists in the field may find the results interesting.
Specific comments:
Line 91 - I assume "anti-anxiety medications" includes SSRIs and SNRIs. It would be best if the authors could be more specific.
Line 94- Likewise, it would be nice to know what sorts of psychotherapies were prescribed.
Table 1 - this seems like two tables in one. The top portion is baseline characteristics and the bottom part reports the main results. At least for the top part, the p-values are meaningless, there is no hypothesis being tested here nor is there any randomization. I would recommend splitting this table into two.
Table-1 - Are the +/- signs signifying standard deviation or standard error?
Statistical analysis - The field has long moved away from reporting p-values as a measure of significance of a finding. Since the main outcome variable here is the difference in mental fatigue between OCPD and control groups, modern practice would be to report the difference, and the CI.
Line 153- I am not sure I understand this finding. An OR is reported but I could not understand "the odds of what" exactly. Please explain.
Author Response
We are grateful for the comments.
Please see the attachment.

Reviewer 2 Report
The report by Gecaite-Stonciene and colleagues aimed to explore the role of fatigue and cognitive functions in patients with comorbid obsessive compulsive personality disorder (OCPD) and mood disorders/anxiety disorders.
Research on OCPD is scant, so the topic is of relevance. I like the theoretical considerations of the study, the manuscript is well written. However, I detect statistical shortcomings resulting in an over interpretation of the results warranting revision.
Title
The title of the paper refers to mental fatigue only whereas the introduction authors suppose a relationship between OCPD traits and fatigue in general. I suppose a more catchy title comprising the main finding of the paper.
Abstract
The abstract should state that only an association between OCPD and mental fatigue but not general or physical fatigue or the other subscales of the MFI-20 were found.
Line 18: mean age should be given in years ;
Line 22, 23: it should say “there was a significant link...“;
Introduction
In the introduction authors state twice that OCPD is frequently comorbid with mood/anxiety disorders which is redundant. Readers should be informed about the frequency of co-occurrence in numbers though. In the presented study sample authors found roughly 23% OCPD which would be very much in line with what we know.
Ref 26 refers to late life cognitive impairment in GAD which makes it rather inappropriate in this context, please provide an alternative citation.
The sentence given in lines 68-70 is not clear to me, how is fatigue after myocardial infarction linked to OCPD? Please clarify.
Methods
A more recent development in the classification of OCPD is the inclusion of two sets of diagnostic criteria in the DSM-5, the official and the alternative set. I recommend a short comment on that fact, also in light of the diagnosis of OCPD made in the current study according to the CPAS.
Regarding statistics, results require Bonferroni-Holm correction for multiple comparisons which leads to a loss of statistical significance also for the item mental fatigue.
Discussion
Line 157: the hypothesis of the authors according to the introduction was an association between OCPD and fatigue, not just mental fatigue; this should be clearly stated and dealt with in the discussion section.
As mentioned earlier, statistical significance is only present in the BINLOG not the t-test which should be accounted for properly in the discussion.
Authors should explain the concept of „mental fatigue“ in a more detailed way.
Please extend the discussion with considerations/thoughts why cognitive functions in patients with a primary OCPD diagnosis seem to be affected compared to healthy controls whereas in patients with psychiatric/neurological disorders other that OCPD as primary diagnosis, like in the present study, comorbid OCPD has no effect on cognitive functions.
Author Response
We are thankful for your time and efforts put in reviewing our manuscript.
Please see the attachment.

Round 2
Reviewer 2 Report
The quality of manuscript has markedly improved, it is suitable for publication in its present from.